# Disease-associated genotypes of the commensal skin bacterium *Staphylococcus epidermidis*

Guillaume Méric[1], Leonardos Mageiros[1,2], Johan Pensar[3], Maisem Laabei[1,20], Koji Yahara[4], Ben Pascoe [1,5], Nattinee Kittiwan[6], Phacharaporn Tadee[7], Virginia Post[8], Sarah Lamble[9], Rory Bowden [9], James E. Bray [10], Mario Morgenstern[11], Keith A. Jolley [10], Martin C.J. Maiden [10], Edward J. Feil [1], Xavier Didelot[12], Maria Miragaia[13], Herminia de Lencastre[13,14], T. Fintan Moriarty [8], Holger Rohde[15], Ruth Massey[1,16], Dietrich Mack[17], Jukka Corander [3,18,19] & Samuel K. Sheppard[1,5,10]

Some of the most common infectious diseases are caused by bacteria that naturally colonise humans asymptomatically. Combating these opportunistic pathogens requires an understanding of the traits that differentiate infecting strains from harmless relatives. *Staphylococcus epidermidis* is carried asymptomatically on the skin and mucous membranes of virtually all humans but is a major cause of nosocomial infection associated with invasive procedures. Here we address the underlying evolutionary mechanisms of opportunistic pathogenicity by combining pangenome-wide association studies and laboratory microbiology to compare *S. epidermidis* from bloodstream and wound infections and asymptomatic carriage. We identify 61 genes containing infection-associated genetic elements (k-mers) that correlate with in vitro variation in known pathogenicity traits (biofilm formation, cell toxicity, interleukin-8 production, methicillin resistance). Horizontal gene transfer spreads these elements, allowing divergent clones to cause infection. Finally, Random Forest model prediction of disease status (carriage vs. infection) identifies pathogenicity elements in 415 *S. epidermidis* isolates with 80% accuracy, demonstrating the potential for identifying risk genotypes pre-operatively.

[1] The Milner Centre for Evolution, University of Bath, Claverton Down Bath BA2 7AY, UK. [2] Swansea University Medical School, Swansea University, Singleton Campus Swansea SA2 8PP, UK. [3] Department of Mathematics and Statistics, University of Helsinki, Helsinki 00100, Finland. [4] Antimicrobial Resistance Research Center, National Institute of Infectious Diseases, Tokyo 162-8640, Japan. [5] MRC Cloud-based Infrastructure for Microbial Bioinformatics (CLIMB) Consortium, Bath BA2 7AY, UK. [6] Integrative Research Centre for Veterinary Preventive Medicine, Faculty of Veterinary Medicine, Chiang Mai University, Chiang Mai 50200, Thailand. [7] Graduate School, Maejo University, Chiang Mai 50290, Thailand. [8] AO Research Institute Davos, Davos 7270, Switzerland. [9] Wellcome Centre for Human Genetics, University of Oxford, Oxford OX3 7BN, UK. [10] Department of Zoology, University of Oxford, Oxford OX1 3SZ, UK. [11] Department of Orthopaedic Surgery and Traumatology, University Hospital Basel, Basel 4031, Switzerland. [12] Department of Infectious Disease Epidemiology, Imperial College, London SW7 2AZ, UK. [13] Laboratory of Molecular Genetics, Instituto de Tecnologia Química e Biológica, Universidade Nova de Lisboa, Oeiras 2775-412, Portugal. [14] Laboratory of Microbiology and Infectious Diseases, The Rockefeller University, New York, New York 10065, USA. [15] Institut für Medizinische Mikrobiologie, Virologie & Hygiene, Universität Hamburg, Hamburg 20246, Germany. [16] School of Cellular and Molecular Medicine, University of Bristol, Bristol BS8 1TD, UK. [17] Bioscientia Labor Ingelheim, Institut für Medizinische Diagnostik GmbH, Ingelheim 55218, Germany. [18] Department of Biostatistics, University of Oslo, Oslo 0372, Norway. [19] Pathogen Genomics, Wellcome Trust Sanger Institute, Hinxton CB10 1SA, UK. [20]Present address: Medical Protein Chemistry, Department of Translational Medicine, Lund University, Malmö 205 02, Sweden. These authors contributed equally: Guillaume Meric, Leonardos Mageiros. Correspondence and requests for materials should be addressed to S.K.S. (email: s.k.sheppard@bath.ac.uk)

The most commonly cultured bacteria in clinical microbiology laboratories are the coagulase-negative staphylococci (CoNS), especially *Staphylococcus epidermidis*[1,2]. Despite their importance as nosocomial pathogens[3–7], CoNS are not routinely surveyed even though they can represent > 40% of cultured isolates from blood or cerebrospinal fluid samples[4,5]. The reasons for this underreporting, compared with notorious nosocomial pathogens such as *Clostridium difficile*[8] and *S. aureus*[9,10], are linked to the ubiquity of CoNS as commensal colonizers of human skin and mucus membranes. This leads to difficulties in determining the clinical significance of isolates for two reasons. First, isolates that have caused infection during invasive procedures[11] are difficult to distinguish from those that have contaminated microbiological samples, unless they are isolated multiple times[12]. Second, the incomplete understanding of the determinants of pathogenicity in CoNS means they are often described as opportunistic or accidental pathogens[13] with little attention given to the emergence and spread of virulent lineages.

There is mounting evidence that *S. epidermidis* isolated from infections are a subset of those found on the skin surface[14–18]. This implies that, rather than simple passive infection, there may be certain lineages or specific virulence factors associated with the emergence of pathogens from a background of harmless ancestors. For example, *S. epidermidis* pathogenesis is associated with antibiotic resistance, attachment to host tissues and accumulation in multi-layered biofilms on implanted medical devices[3,19,20]. Consistent with this, methicillin resistance (*mecA*) and virulence genes known to encode polysaccharide intercellular proteins (PIA) are over-represented among strains from clinical samples[15,16,21].

With ever-more reliance upon invasive surgery in the post antibiotic era, device-associated infections caused by *S. epidermidis* will increase[22]. Therefore, there is a pressing need to monitor these bacteria within genetically diverse commensal populations[3,23,24], and identify strains that may be pre-disposed to pathogenicity. Here, we identify genetic and functional traits associated with pathogenicity among 415 *S. epidermidis* isolate genomes from asymptomatic carriage and human disease. Applying a genome-wide association study (GWAS) approach linked to clinically relevant phenotypes tested in vitro, we identify whole genes and genetic elements associated with pathogenicity (Fig. 1). This study improves the understanding of the evolution of virulence and allows the calculation of a risk score for individual isolate genotypes that, with further validation, could be a basis for medical interventions.

## Results

### Core and accessory genome variation in *S. epidermidis*.
The pangenome of the 415 *S. epidermidis* isolate dataset comprised 12,079 unique genes. These included 1946 genes present in all isolates which corresponded to 72% of the average genome size, consistent with previous core genome estimates[24]. The rate of accessory gene discovery did not plateau as the sampling increased (Supplementary Figure 2), suggesting widespread acquisition of genes through horizontal gene transfer (HGT). While only 36% of all annotated genes from the reference *S. epidermidis* strain ATCC12228 were of unknown function, this number increased to 72% for the whole pangenome. All the assembled genomes analysed in this study are available via Figshare (https://doi.org/10.6084/m9.figshare.7058543).

### Pathogenicity emerges from asymptomatic lineages.
The population structure of 415 *S. epidermidis* isolates from infection and asymptomatic carriage was reconstructed using a maximum-likelihood phylogenetic tree from a concatenated gene-by-gene alignment of 1946 genes shared by all isolates (Fig. 2b). Topology was consistent with previous studies[23,24] demonstrating that the isolates in our collection represented known population structure within the species. Infection isolates were present across the tree reflecting emergence of disease clones from multiple genetic backgrounds[23] (Fig. 2b). A total of 355 isolates corresponded to 82 different sequence types (STs) (Table S1, Fig. 2a), with > 60% of isolates (254/415) clustered in a single clonal complex (CC-2). It remains possible that clonal lineages with enhanced pathogenic potential may exist somewhere, or emerge in the future, but among known genetic diversity in this species, isolates from all major phylogenetic groups were represented in both the asymptomatic and the infection isolate collections.

### Pan-genome-wide association study of infection-associated genes.
Replicate GWAS experiments were performed on two datasets of paired isolates with high sequence identity but divergent phenotypes (asymptomatic carriage vs. infection) (Supplementary Data 2, Supplementary Figure 6). This reduced the impact of population structure and maximised the chance of identifying elements associated with a phenotypic switch. A total of 231,895 and 709,439 associated k-mers, respectively, mapped to 914 and 1320 unique genes in the reference pan genome for an overall total of 54,244 distinct alleles. There were a total of 636 genes containing infection-associated k-mers in both replicate GWAS runs (Fig. 2c, Supplementary Data 3), corresponding to 250 core and 386 accessory genes. These genes had diverse predicted functions including those involved in toxicity, adhesion, biofilm formation and metabolism, consistent with multifactorial pathogenicity (Supplementary Data 4, Supplementary Figure 3). Nearly half (17/40) of the top 40 genes containing significantly associated k-mers were components of the staphylococcal cassette chromosome mec (SCC*mec*) cassette (Supplementary Data 3). As in GWAS studies of other organisms[25,26], these candidate associations have potential to improve understanding of known and novel factors related to infection.

### Correlating pathogenicity k-mers and in vitro phenotypes.
The prevalence of associated k-mers from primary GWAS (carriage vs. infection) was correlated with quantitative scores from laboratory phenotype assays, related to staphylococcal pathogenicity (Fig. 1). While all hits from the primary GWAS have potential for use as infection biomarkers, this correlation step places putative genomic associations in the context of established bench-top microbiology allowing functional inference and improved understanding of clinically relevant genotype–phenotype associations. Laboratory phenotypes included biofilm formation[27], methicillin resistance[28], cell toxicity[29,30] and post-infection interleukin-8 (IL-8) levels in skin epithelial cells and blood serum[31,32] (See Supplementary Methods, Fig. 3a–e, Supplementary Figure 4, Supplementary Data 6). We observed no significant phenotype differences between asymptomatic carriage and infection strains for IL-8 production in keratinocytes ($p = 0.2617$, two-tailed $t$ test, t = 1.141, df = 35) (Fig. 3a), or biofilm formation ($p = 0.0856$, two-tailed unpaired $t$ test, t = 1.741, df = 78) (Fig. 3b). However, there was a general trend towards increased methicillin resistance ($p < 0.0001$, two-tailed Fisher's exact test) and reduced toxicity ($p = 0.0188$, two-tailed unpaired $t$ test, t = 2.435, df = 46) among infection isolates. This is consistent with previous studies of methicillin resistance among clinical isolates[15] and lower cell toxicity among isolates from invasive disease[33], highlighting a role for the reduction of cytolytic activity to be a favourable trait for relative fitness in human serum[34]. Genetic variation in *agrC*, associated with a single k-mer hit (Supplementary Data 1), may confer attenuated

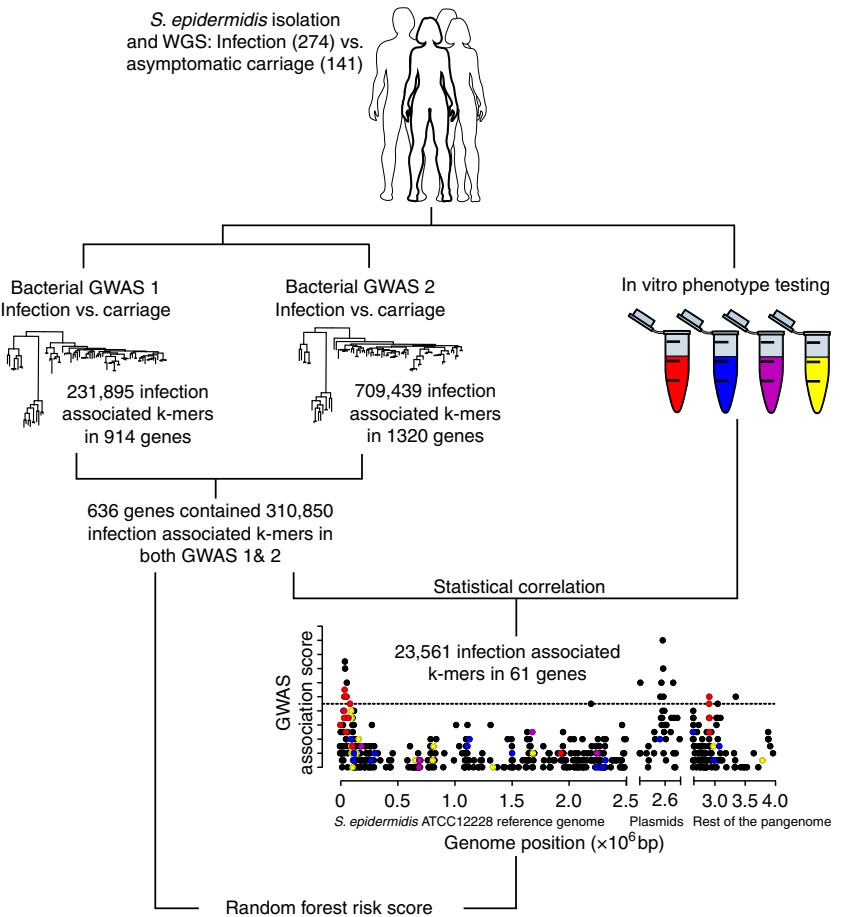

**Fig. 1** Phenotype correlated GWAS and risk prediction. Genome-wide association studies (GWAS) can identify numerous SNPs associated with complex traits but these can be difficult to interpret. For example, pathogenicity is multifactorial, potentially involving genes underlying phenotypes that promote transmission, virulence, immune evasion, antimicrobial resistance etc. In some bacteria, specific phenotypes known to contribute to pathogenicity can be measured in laboratory assays, providing a basis for quantitative analysis of disease dynamics. We developed a method in which k-mers from a primary GWAS analysis (asymptomatic carriage vs. infection isolates) were correlated with data from four relevant phenotype assays: biofilm formation (blue); cell toxicity (yellow); methicillin resistance (red); IL-8 production by host cells (purple). Using Fisher's exact test, k-mers from primary GWAS were correlated with laboratory phenotype using a 2 × 2 table in which rows indicated presence/absence of the k-mer and columns indicated upper and lower percentile in the laboratory phenotype assay. The resulting *P*-values, derived for each k-mer, help to link k-mers in the primary GWAS with quantifiable pathogenicity-related phenotypes. Patterns of k-mer presence and absence can be used as classifiers in a random forest model to identify the best predictors of infection

function of AgrC[35], associated with persistent *S. aureus* bacteraemia[36]. We observed a significantly higher production of IL-8 in blood in infection compared with carriage isolates ($p = 0.0185$, two-tailed unpaired *t* test, t = 2.405, df = 78) (Fig. 3d). The prevalence of all k-mers from the primary phenotype was correlated (Fisher's exact test) with isolate phenotype variation in the laboratory assay (Fig. 1). A total of 23,561 (out of 210296, Supplementary Data 3) pathogenicity-associated k-mers correlated with high in vitro phenotype scores, corresponding to 61 genes: 17 involved in biofilm formation; 18 in cell toxicity; 8 in IL-8 response to infection in blood; 18 in methicillin resistance (Supplementary Data 1, Fig. 3f, Supplementary Notes). The frequency of these 23,561 correlated k-mers was quantified in a second dataset of 263 *S. epidermidis* genomes (Supplementary Data 2), comprising 65 carriage and 198 infection isolates. The presence of a given infection-associated k-mer was strongly predictive of the presence of other k-mers associated with that secondary phenotype (Supplementary Figure 5). A total of 3% of

carriage isolates contained infection-associated elements for all four laboratory phenotypes combined, compared with 58% for infection isolates.

**Consistency index of pathogenicity-associated genes**. There was a subtle increase in the average allelic variation among genes associated with infection (Fig. 4a). This might be expected because of the accumulation of deleterious mutations associated with bacterial range expansion[37], but the difference was not statistically significant. The average number of unique alleles per isolate was $0.2442 \pm 0.1494$ for the 61 genes containing phenotype correlated infection-associated elements, compared with $0.1415 \pm 0.065$ for 1946 core genes. The different distributions of these values (Fig. 4b) provided an initial indication of elevated recombination among pathogenicity-associated genes. Consistent with this, the mean consistency index (CI) was significantly lower (Mann–Whitney test; $U = 15.50$, $p = 0.002$) among genes

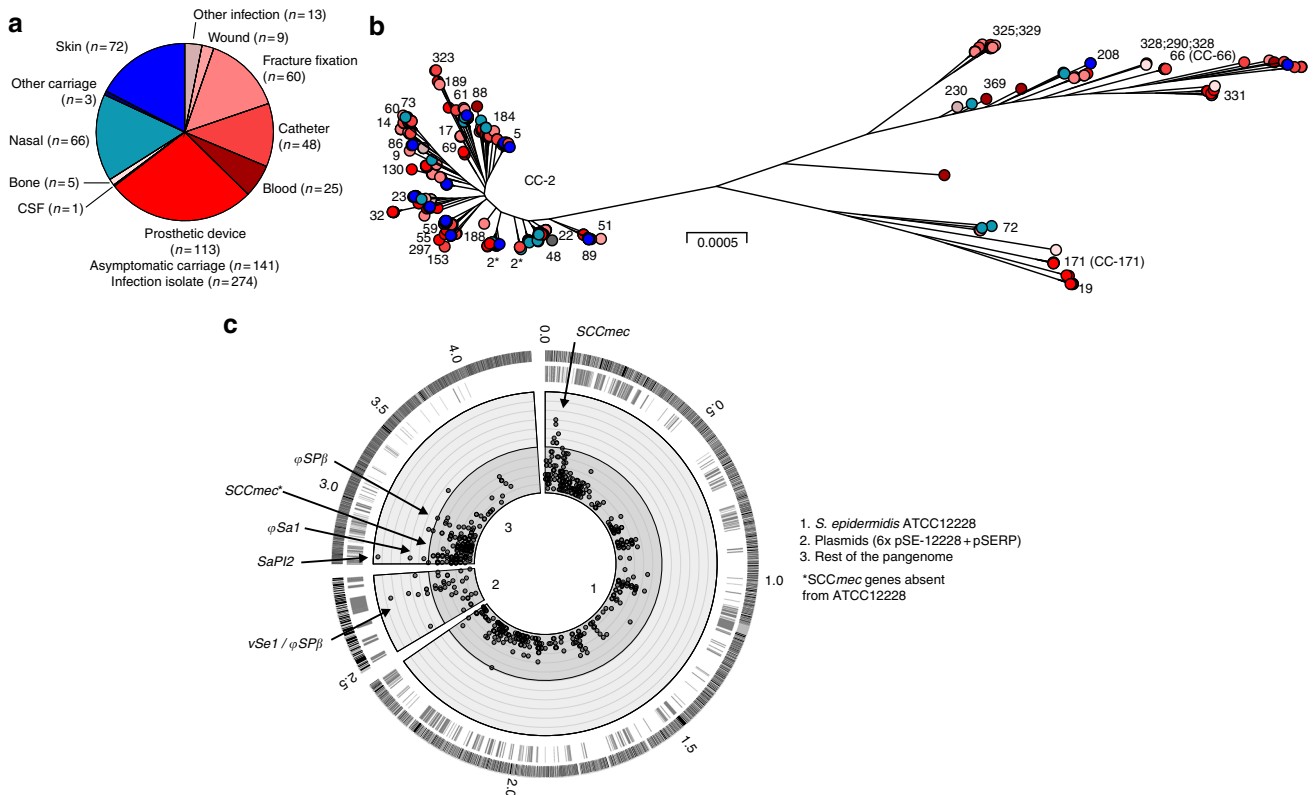

**Fig. 2** Population structure and genome-wide association study of *S. epidermidis* pathogenicity. **a** Isolation source of 274 infection and 141 asymptomatic carriage isolates. Shades of red correspond to the broad infection phenotype, shades of blue to the broad asymptomatic carriage phenotype. **b** Phylogenetic tree of 415 *S. epidermidis* isolates, reconstructed using an approximation of the maximum-likelihood algorithm (FastTree2) from a core genome alignment (n = 1946 genes shared by all isolates). Colours correspond to those used in panel A. Numbers correspond to sequence-types (STs) according to the Miragaia MLST scheme (https://pubmlst.org/sepidermidis/). **c** Pangenomic position of GWAS results. Ticks in the outer ring represent the pangenomic position of genes in the *S. epidermidis* ATCC12228 reference genome, seven plasmid genomes, and the rest of the pangenome inferred in this study. Ticks in the second layer show the position of genes containing associated k-mers in the GWAS. Grey circles are shown for the most statistically associated (lowest p-value) k-mer mapping to that gene. The threshold for significance (Fisher exact test) was P-value = 0.0001 (inner circle). Concentric rings emanating from this threshold correspond to incremental reductions in *P*-values from $1 \times 10^{-4}$ to $5 \times 10^{-7}$ (dark grey rings) and values $< 5 \times 10^{-7}$ (light grey rings). The position of genes in well-known pathogenicity islands (SCCmec, φSPβ, φSa1, SaPI2, vSe1/ φSPβ) is indicated

containing infection-associated elements (0.3064 ± 0.21) compared with other core genes (0.4590 ± 0.1577), and the respective distributions of all CI values clearly differed (Fig. 4b). This provides evidence that the clonal mode of descent is disrupted in infection-associated genes consistent with elevated HGT.

**Infection risk genotypes.** Quantitative determination of markers of infection risk was carried out using a Random Forest (RF) approach in three ways where the estimated risk score was defined as: (i) the probability of an isolate coming from infection given a certain k-mer profile; (ii) the probability of an isolate coming from infection given a certain phenotype correlated k-mer profile; (iii) the probability of an isolate coming from infection based upon the presence of the four k-mers that were identified as most important for each of the four clinically relevant lab phenotypes (Fig. 1). In the initial RF analyses (i) and (ii), all 1900 and 293 (respectively) unique k-mer presence and absence patterns were included as predictors in the model. The models reached out-of-sample classification accuracy of 85.4% and 80.5%, respectively, for predicting disease status (infection vs. carriage) based on the k-mer profile. K-mers associated with SCCmec accounted for a high proportion of the most important predictors, with five in the top ten (Fig. 3g). To investigate the amount of redundancy among the k-mer predictors, they were sorted according to their estimated importance and sub-models

including only the *l* most important phenotype correlated predictors (*l* = 1,…,293) were built and evaluated. The importance of the 20 highest ranked predictors is shown in Fig. 3h alongside the classification accuracy of the corresponding sub-models. There was considerable redundancy among the predictors. The classification accuracy of most sub-models was around 80%, the highest ranked MEC-associated predictor reached a classification accuracy of around 75% on its own, potentially offering a very simple target for clinical investigation of *S. epidermidis* risk.

Given the results in studies (i) and (ii), information provided by the k-mers can be captured almost fully using a much simpler model. Using the ranking provided by the initial studies, the final model (iii) was built using only the most important predictor from each lab phenotype category. The selected predictors were found on place 1, 2, 3 and 11 in the importance ranking for phenotype-correlated predictors (Fig. 3h). The significantly simpler model with only four predictors reached an out-of-bag classification accuracy of 79.8%, which is close to that in the complete model. The importance of the selected predictors in the new model is shown in Fig. 3I. The high numbers of SCCmec-associated elements in the primary GWAS (Supplementary Data 3) and among the 61 genes containing phenotypically correlated hits (including *ccrB, mecR1, mecA, maoC, arc, arcB-2* and other genes encoding hypothetical proteins, Supplementary Data 1) indicates the importance of relative abundance of

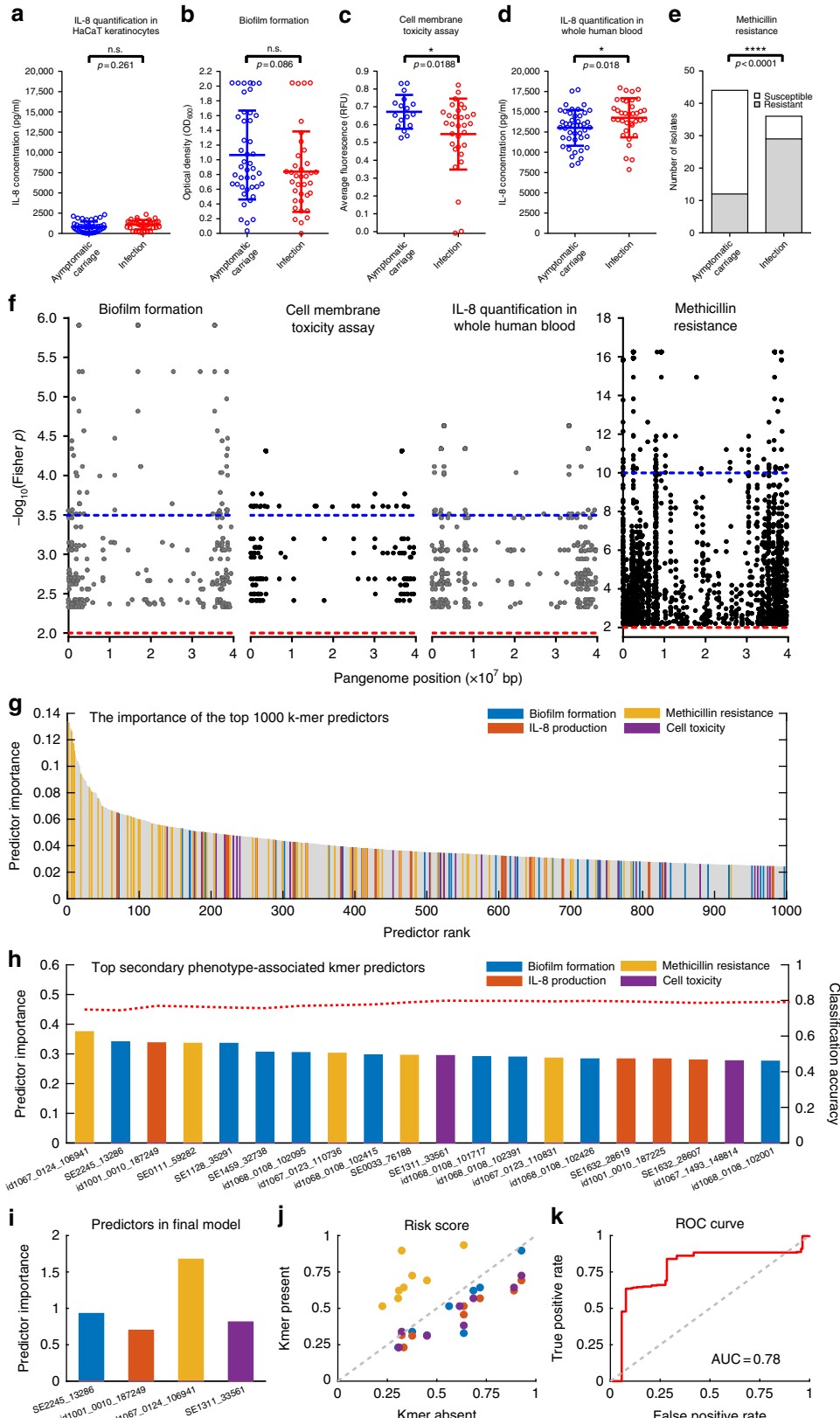

SCCmec in infection strains, compared with those from the commensal environment. Consistent with this, the MEC-associated predictor, in *mecA*, was clearly the most important. Figure 3j illustrates the overall effect of the different k-mers on the estimated risk score of the model. A point above the diagonal implies that the risk score for a specific k-mer profile is higher when the colour-indicated k-mer is present compared with absent. Overall, the presence of the MEC-associated k-mer significantly increases the risk score, while the presence of the other k-mers has a more moderate opposite effect. Finally, to

**Fig. 3** Correlation of pathogenicity-associated genotypes with in vitro pathogenicity-related phenotypes. **a–e** Distribution of scores for five in vitro phenotypes between asymptomatic carriage (red, n = 44) and infection S. epidermidis isolates (blue, n = 36): **a** interleukin-8 (IL-8) quantification in HaCaT keratinocytes and **d** in whole human blood serum, **b** biofilm formation, **c** cytotoxicity using a vesicle assay (comparing 17 asymptomatic with 31 infection isolates), and **e** methicillin resistance (defined as growth at a concentration of > 0.25 mg/L). The mean and s.d. error bars are shown for A-D with P-value determined using two tailed t test, n.s. indicates not significant. Two-tailed Fishers exact test was used to determine significant difference (P-value) in methicillin resistance (**e**). **f** Manhattan plot of Fisher's exact test P-values correlating the prevalence of each GWAS-associated k-mer with high and low percentiles of four in vitro phenotype scores performed on the same S. epidermidis isolates used for GWAS. The red dotted line indicates the lower threshold for statistical significance used in the GWAS. The blue dotted line indicates a cut-off for top correlation values. Top values mapped to 61 genes. **g–k** Identification of predictive genotypes for pathogenicity in S. epidermidis using random forest (RF) models. **g** Importance of the top 1000 (of 1900) k-mer predictors from the primary GWAS; **h** predictor importance (left y-axis) among the top 20 phenotype correlated predictors. The red dotted line shows the classification accuracy (right y-axis) of the sub-models in which only the corresponding top predictors are included. **i** Predictor importance of the four laboratory phenotype-specific k-mers included in the final model. **j** Change in risk score for a specific k-mer profile when the colour-indicated k-mer is present (y-axis) compared to absent (x-axis). A point above the diagonal implies that the risk score is increased when the k-mer is present. **k** ROC curve showing the overall performance of the classifier

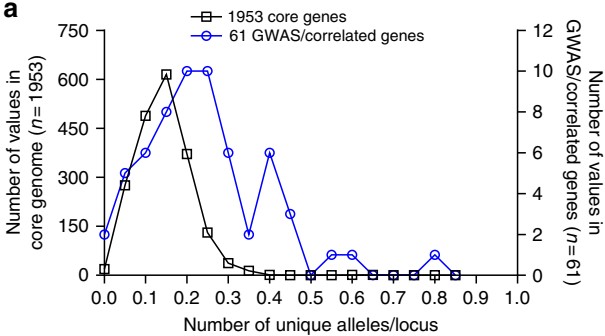

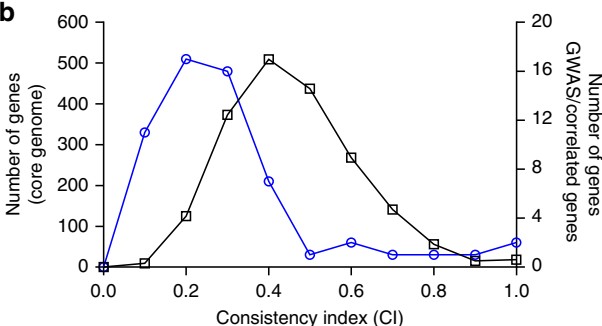

**Fig. 4** Comparison of allelic variation and consistency index for core genes and genes containing GWAS pathogenicity elements. **a** The number of alleles per locus and **b** consistency indices to a core phylogeny, were calculated for each gene alignment for core genes and those containing pathogenicity-associated elements that correlated with secondary in vitro phenotypes using R and the *phangorn* package. The left y-axis indicates the number of core genes (black line), the right y-axis indicates the number of genes containing associated and correlated k-mers (blue line). For the consistency index, the two distributions were significantly different (two-tailed Mann–Whitney test; P = 0.002, Mann–Whitney U = 15.50)

illustrate the trade-off between true and false positives, the ROC curve based on the out-of-bag risk scores of the classifier is shown in Fig. 3k.

The greatest challenge for risk prediction based upon infection-associated k-mers is that samples from asymptomatic carriage may include strains that have the potential to cause infection later, after our samples were taken. This depends on the opportunity to infect, specifically the healthcare related procedures a person will be subjected to. Thus, while it is relatively straight forward to obtain isolates from confirmed infection, it is nearly impossible to get a representative sample of carriage strains that does not contain isolates with the potential to cause disease.

In light of this, it may seem surprising that a single k-mer classifier can be so powerful. This has clear implications for the development of infection biomarkers in a clinical setting. Finally, we carried out an additional validation RF analysis using the best RF classifier k-mer (in *mecA*), on a small independent dataset of S. epidermidis isolate genomes comprising 18 commensal carriage and 18 randomly selected infection isolates (from 312) available on the NCBI database (Supplementary Data 7). The classification accuracy was 67%, which is comparable with that in the larger primary dataset (75%).

## Discussion

Many infections are caused by pathogens that arise from a background bacterial population that, under normal circumstances, co-exists peacefully with their hosts[38]. For bacteria, infection requires the opportunity for transmission and the ability to proliferate in the infection niche. In nosocomial staphylococcal disease, transmission is primarily a passive process as commensal epidermal organisms infect under conditions of host perturbation, for example, through contamination of subcutaneous tissue during invasive procedures. However, disease also depends on pathogen survival and colonization of the new subcutaneous niche, where environmental conditions are different.

There is compartmentalization of the environments from which S. epidermidis was sampled in this study into strains from commensal carriage (skin, nasal pharynx) and infection. It is therefore possible to consider different possible models for S. epidermidis infection from the primary site of adaptation (commensal niche) when there is epidermal damage (Fig. 5). First, the proliferation of specific pathogenic clones that are a sub-population of the commensal skin microbiota. Second, true opportunistic pathogenicity, in which all strains are equally able to cause infection. Third, a divided genome model[39–41], in which strains from multiple genetic backgrounds proliferate because they share genes and associated phenotypes that promote colonization of the subcutaneous niche.

In a simple infection model, disease may result from the bacteria adapting to one or few dominant ecological changes, such as resistance to antibiotics that may be abundant in the tissue of hospital patients. S. aureus provides a good example, as progenitor strains have acquired resistance through rare HGT events and the descendants proliferate because of the advantage this provides in the invasive niche. In cases such as this, it may be possible to identify the spread of successful pathogen clones (Fig. 5a) by comparing them with commensal isolates on phylogenetic reconstructions, where they appear as clusters of genetically related disease-causing strains[42].

It is clear from the S. epidermidis phylogeny (Fig. 2b) that disease isolates do not represent a few successful pathogen clones,

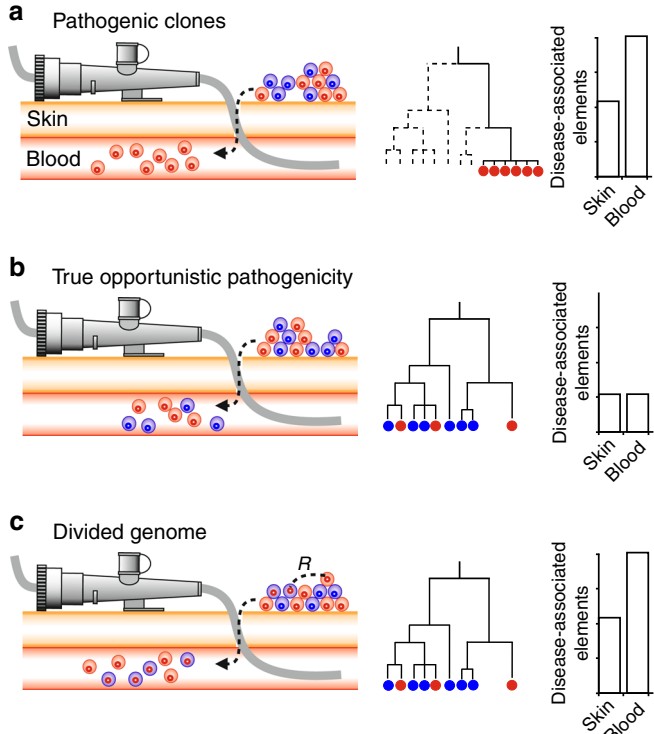

**Fig. 5** Contrasting models of *S. epidermidis* infection and associated variation in conceptual genomic data. Each panel summarises scenarios for subcutaneous colonization from the primary commensal skin environment to the blood (left), and the impact on an *S. epidermidis* population of two clones (blue and red circles) and their genomes (internal circles) which may be enriched for putative pathogenicity-associated genes (red) or not (blue). Genealogical reconstructions of isolates sampled from infected blood are shown in the middle column. The prevalence of disease determinants in the genome of isolates from skin and blood are shown on the right. **a** Proliferation of pathogenic clones: clones with genomes enriched for pathogenicity determinants proliferate in the blood and other strains do not, observed as a discrete pathogen lineage on the tree. **b** True opportunistic pathogenicity: multiple genetically divergent clones proliferate in the blood and disease determinants are equally distributed among the genomes of isolates from the skin and the blood (or would be undetectable). **c** Divided genomes: horizontal gene transfer (R) spreads pathogenicity determinants into multiple genomic backgrounds allowing divergent clones to colonize the blood successfully

but are distributed across the phylogeny with commensal isolates of multiple genetic backgrounds. This could imply that all *S. epidermidis* lineages are equally able to cause infection, given the opportunity for transmission (Fig. 5b). If this were the case, then disease determinants would not be expected to segregate by isolate source. However, the GWAS identified numerous infection-associated k-mers, many of which mapped to genes known to be associated with pathogenicity[16]. This is consistent with enrichment for sequence encoding traits such as colonisation, survival and virulence factors among isolates from infection.

To investigate putative functional differences between invasive and commensal strains, it is necessary to link infection-associated SNPs with phenotypic variation. This can be challenging when there are high numbers of infection-associated k-mers, reflecting the multifactorial nature of the pathogenicity. Some genetic variation can relate to more than one type of infection[43], but by correlating k-mer presence with laboratory phenotypes (Fig. 1) known to be relevant to staphylococcal virulence[28–30,32], we identified sequence variation in *S. epidermidis* genes associated

with biofilm formation, cell toxicity, methicillin resistance and elicitation of inflammation in blood (IL-8). The enrichment of these putative virulence determinants among isolates from infection suggests that pathogenic strains are a subset of the commensal population that contain genes and alleles that may promote colonization of the site of infection (Fig. 5c).

This is difficult to explain from a theoretical point of view as the commensal environment is the primary niche and isolates encounter the secondary (invasion) niche relatively infrequently. Therefore, there is little opportunity for an evolutionary trade-off between genes that favour growth in one niche versus the other[38], and the fitness of pioneer populations may be expected to decline as they expand their range because of increased genetic drift and reduced efficiency of selection in removing deleterious mutations[37]. Furthermore, while chronic infection could represent a reservoir for re-colonization of the skin, in many cases the secondary niche will be a dead end, especially as the host patient may be cured or die. This means that adaptations to the secondary niche would be purged from the population because of their fitness cost. It is possible that virulence-associated variation confers a different advantage in the primary niche. This type of pre-adaptation has been observed in *Streptococcus pneumoniae* where selection for maintenance of capsular polysaccharides is driven by competition with other commensal organisms in the nasal pharynx but also confers increased risk of causing invasive disease in humans[44,45].

While pre-adaptation may be important, HGT is known to be a major force in staphylococcal evolution, including *S. epidermidis*[24], with the acquisition of genes through recombination potentially conferring adaptations associated with pathogenicity[24,46]. This has potential benefits in heterogeneous environments[47], extending the number of niches that *S. epidermidis* can colonize successfully, through the acquisition of virulence factors that promote proliferation on invasion. Initial evidence of HGT can be seen as the putative virulence determinants identified with GWAS, are not distributed consistent with the *S. epidermidis* clonal frame (Fig. 2b) and it is unlikely that convergent genotypes evolved multiple times in different genetic backgrounds. Furthermore, detailed analysis of individual trees revealed that the 61 genes containing putative virulence determinants have a significantly lower mean consistency index (Fig. 4b), compared with core genes, suggesting more homoplasy has occurred.

While clonal reproduction might be expected to dominate in the primary commensal niche the importance of HGT may be elevated in heterogeneous environments allowing adaptive genetic elements to spread horizontally through the population. This is consistent with a divided genome[39] or gene-specific selective sweep model[40,41] of bacterial evolution where genes, rather than strains, inhabit niches. When there is migration, the rate and impact of HGT would be increased[39] as genes that are positively selected in the infection niche will sweep through the population. Recombination, therefore, increases the speed and effectiveness of adaptation to the invasive niche by ameliorating competition between selected clones carrying competing beneficial mutations (clonal interference[48]) by moving multiple selected sites into a common background (Fig. 5c). Therefore, in a genetically diverse community of commensal *S. epidermidis*, HGT may promote: (i) the emergence of lineages (at the boundary between niches) that could colonize the invasive niche effectively and (ii) ongoing adaptation as positively selected genes that confer an advantage in the invasive niche, sweep through the invasive population.

Finally, we turn to the control of *S. epidermidis* infection in a public health setting. Based upon the findings of this study, it is clear that targeting individual clones based upon molecular typing methods would be partially ineffective, as the putative

determinants of disease recombine and are found in multiple genetic backgrounds. However, predicting the likelihood that a given isolate from asymptomatic carriage could lead to complications after surgery would be of benefit, allowing pre-operative interventions to reduce the risk of infection. Simple empirical comparison of the frequency of putative virulence determinants allows the identification of carriage strains that may pose a risk (Supplementary Figure 5). This has limitations as there are numerous colonization and virulence factors that may be associated with different types of infection, for example bacteraemia versus indwelling device infection, and not all virulence factors will necessarily be present among *S. epidermidis* isolates causing infection[49].

To provide a more accurate prediction of risk, we used a Random Forest machine learning approach to quantify the power of different k-mer combinations to predict if an isolate came from infection. Based upon several analyses, there was considerable redundancy among the predictors, with little benefit in analysing all disease-associated k-mer combinations. Using a simple model with only the most important k-mer predictors from each laboratory phenotype category (Fig. 3h) gave a classification accuracy of 80%. Interestingly, a k-mer mapping to the *mecA* gene, that encodes methicillin resistance on the SCCmec element, was the best predictor for *S. epidermidis* isolates from infection, giving a classification accuracy of 75% on it's own. It is well known that Methicillin-resistant *S. aureus* (MRSA), harbouring SCCmec, are a prominent cause of infection in healthcare settings[9]. Furthermore, the presence of the *mecA* locus is also correlated with resistance to fluoroquinolones[50,51], associated with point mutations in *grlA, grlB, gyrA* and *gyrB* genes and *gyrB* was also among the 61 genes containing k-mers correlated with *S. epidermidis* methicillin resistance in vitro (Supplementary Data 1). These parallels may suggest that methicillin resistance has a similar role in the epidemic potential of *S. aureus* and *S. epidermidis*. However, the epidemic spread of MRSA associated with a discrete number of highly successful MRSA clones[9] contrasts with the emergence of multiple disease-causing *S. epidermidis* clones distributed across the phylogeny (Fig. 2b). Under these conditions, the risk markers have considerable potential for identifying pathogenic strains and, as larger numbers of isolate genomes increase the predictive power, these models could be used for evaluating pre-operative treatment options in a public health setting after further validation.

Opportunistic pathogens, such as *S. pneumoniae, Neisseria meningitidis* and *S. aureus*, remain a major public health threat. These organisms do not conform to a simple theoretical closed system model of obligate pathogen specialists, but clues to the factors that promote the emergence of disease-causing strains can be locked in their genome. In the extreme case of *S. epidermidis*, practical difficulties in defining disease-causing strains have led to its underrepresentation among nosocomial pathogen surveys. However, *S. epidermidis* strains from the commensal environment and disease are not equivalent. Rather, the disease-causing *S. epidermidis* represent a pathogenic sub-population that have acquired genetic elements and related phenotypes that promote infection. Defining these organisms as pathogens is the first step towards effective control of infection.

## Methods

**Bacterial sampling**. Genomes from 415 *S. epidermidis* isolates, from multiple sampling efforts, were analysed (Supplementary Data 2). These included: 240 isolates sampled as part of this study; 35 genomes available from public repositories (February 2013); and 140 recently sequenced *S. epidermidis* genomes from geographically and clinically diverse isolates characterised in previous studies[21,23,24,52]. Asymptomatic carriage isolates were sampled from healthy volunteers in Swansea University (UK) in 2012, using culture swabs containing Ames media, which were then cultured on Columbia Blood Agar plates. Volunteers gave informed consent,

as assessed by the local Human Tissue Act committee (Wales REC 6) at the Swansea University Medical School (ref: #13/WA/0190). To ensure that isolates from infection were not laboratory contaminants, 113 strains from prosthetic joint infections isolated from independent pure cultures of pre-operative joint aspirates, and intraoperative tissue specimens were obtained under strict aseptic conditions[16]. Additionally, isolates were sampled from intraoperative surgical specimens of fracture fixations ($n = 60$), osteomyelitis ($n = 5$), bacteraemia ($n = 2$) and colonised catheters linked to an infection event ($n = 45$). Among these, 85 isolates from infection (identifier 1043–1136) were collected as part of a prospective study performed between November 2011 and September 2013 at the BGU Murnau, Germany, a level-one trauma centre with a high volume, 70-bed unit for septic and reconstructive surgery[52,53]. The total dataset in this study comprised 141 isolates from healthy carriage obtained in hospitals and the community from 11 countries in three continents (57/141 from the UK) and 274 isolates from clinical infections (Supplementary Data 2).

**Genomic DNA extraction, sequencing and archiving**. DNA was extracted using the QIAamp DNA Mini Kit (QIAGEN, Crawley, UK), using manufacturer's instructions, with 1.5 µg/µl lysostaphin (Ambi Products LLC, NY) to facilitate cell lysis. DNA was quantified using a Nanodrop spectrophotometer, as well as the Quant-iT DNA Assay Kit (Life Technologies, Paisley, UK). High-throughput genome sequencing was performed using a HiSeq 2500 machine (Illumina, San Diego, CA), and the 100-bp short-read paired-end data were assembled using the de novo assembly algorithm, Velvet[54]. The VelvetOptimiser script (version 2.2.4) was run for all odd k-mer values from 21 to 99. The minimum output contig was size set to 200 bp with the scaffolding option disabled. Other program settings were as default, and assembly quality metrics were recorded (Supplementary Data 5). All genome sequences were archived on a web-accessible BIGSdb database[55], and genome sequences generated in this study are available on NCBI BioProject PRJNA433155.

**Core and accessory genome characterization**. A *S. epidermidis* coding sequence pangenome gene list was constructed for isolates in this study[56] by automated annotation of all genomes from the dataset using the RAST/SEED system[57] and the WebMGA COG annotation server[58] (see Supplementary Methods). After removal of alleles of the same gene with a BLAST threshold of 70% sequence similarity[56], there were 12,079 unique genes present in at least one of the 415 genomes. Consistent with previous studies, and the whole-genome MLST principle[55,59–61], the gene complement and allelic variation of each isolate was determined by comparison with the pangenome with gene presence recorded as a BLAST match of > 70% sequence identity over ≥ 50% of sequence length. For each pair of isolates, the number of shared genes and alleles (identical sequences at a given locus) was calculated. Core genes were present in 100% of the genomes and accessory genes were present in at least one isolate.

**Phylogenetic analyses**. Core gene sequences were individually aligned, using MUSCLE[62], and concatenated, consistent with the gene-by-gene approach[55,60,61]; and a tree was reconstructed using an approximation of maximum-likelihood phylogenetics in FastTree2[63]. This tree was used as an input for ClonalFrameML[64] to produce core genome phylogenies with branch lengths corrected for recombination.

**In vitro phenotype assays**. To measure variation in clinically relevant phenotypes for 80 isolates, established in vitro laboratory assays quantified: (i) biofilm formation; (ii) toxicity using a vesicle lysis test (VLT) (for 48 isolates); (iii) methicillin resistance; (iv) production of interleukin-8 (IL-8) by human keratinocytes in presence of *S. epidermidis*; (v) IL-8 production following inoculation of human blood with *S. epidermidis*. Briefly, biofilm formation was assessed using crystal violet staining of bacteria attached to the polystyrene surface of a 96-well microtitre plate[65], in three biological replicates for each bacterial strain, grown for 24 h at 37 °C in tryptone soy broth (TSB) and washed in PBS (see Supplementary Methods). Methicillin resistance was quantified using standard European Committee on Antimicrobial Susceptibility Testing (EUCAST)[66] methods for susceptibility testing[67]. Bacteria were cultured in the presence of Etest strips (bioMerieux) comprising a pre-determined continuous gradient of methicillin for ~16 h. The minimum inhibitory concentration (MIC) was recorded based upon the zone of inhibition. Cell toxicity was assessed using a vesicle lysis test (VLT)[29] designed to be specific to small amphipathic peptides, including staphylococcal delta and phenol soluble modulin (PSM) toxins (see Supplementary Methods). Briefly, a solution of lipid vesicles containing encapsulated self-quenched fluorescent dye, 5 (6)-carboxyfluorescein (CF), were designed to be responsive to specific *Staphylococcus* toxins so that when the vesicles were disrupted by bacterial supernatants containing secreted cytolytic factors, an increase of fluorescence was measured (see Supplementary Methods)[29]. IL-8 was chosen as an immune response marker from a suite of cytokines as it is known to be important in mediating the pro-inflammatory response in staphylococcal infection, culminating in neutrophil recruitment in pathogen defence. Overexpression of IL-8, along with TNFα, IL-6 and IL-1B, has been postulated as a biomarker for staphylococcal sepsis[68]. IL-8 production by a HaCaT keratinocytes (ATCC)[69] cell line from human skin

epithelial, and by human whole blood was measured by enzyme-linked immuno-sorbent assay (ELISA) after challenge by 80 strains (in three biological replicates) of *S. epidermidis* representing the genomic diversity of the species[23]. These phenotype assays were chosen as they have been previously related to pathogenicity. However, it should be noted that *S. epidermidis* is principally adapted to the commensal niche (17) with no clear virulence-associated phenotype that completely distinguishes invasive from commensal strains[70,71]. The incomplete understanding of pathogenicity means that it is possible that a given phenotype may promote opposite outcomes, for example, infectivity (and acute infection) on one hand and adaptability (chronic infection) on the other. Full details of in vitro phenotype assays are included in Supplementary Methods.

**Pangenome-wide association study**. The alignment-free GWAS method involved fragmentation of assembled genomes into consecutive, overlapping 30 bp k-mers (or "30-mers", termed "k-mers" throughout this study), and sorting by isolate source (asymptomatic carriage vs. infection), capturing genetic variation in the core and accessory genome[59,72]. The prevalence of each k-mer in the two phenotypic groups was quantified in a $2 \times 2$ contingency table (with four cells a, b, c, d) in which rows indicated presence/absence of the k-mer and columns indicated phenotype. Because bacteria reproduce clonally, sequences present in related strains will not only reflect adaptive elements associated with the phenotype of interest, but also sequence that was inherited from the common ancestor, potentially confounding GWAS analysis[25,72–74]. To account for this, two steps were taken. First, duplicate input datasets were defined, each containing 38 unique isolate pairs (one from asymptomatic carriage, one from infection) that are closely related on a ClonalFrameML phylogeny (Supplementary Figure S6). These two datasets were technical replicates for independent GWAS analyses. Second, the significance of the association score (P-value) for each k-mer, $a + d - (b + c)$, was determined by comparing the observed association score with a Monte Carlo simulated null distribution where k-mers where randomly gained and lost along the branches of the clonal phylogeny, independent of the phenotype of interest (asymptomatic carriage vs. infection). Algorithmic comparison of the simulated and observed k-mer score distributions allows correction of the P-values to account for the phylogenetic relationships[72]. Details of the pipeline and scripts are available in Supplementary information (see Supplementary Figure 1) and on https://github.com/shepardlab/pGWAS. To allow functional inference, the significantly associated k-mers (P < 0.001) were mapped to the coding sequence pangenome described above[56], and allele at each locus were identified. The reference pangenome approach is detailed in the Supplementary information.

**Covariance of GWAS hits with secondary in vitro phenotypes**. All k-mers significantly associated with the primary phenotype (asymptomatic carriage and infection) were correlated with data from in vitro phenotypes for that isolate. Results from quantitative biofilm formation, methicillin resistance, cell toxicity and host cell immune response phenotype assays were divided into three categories with a third of ranked values in each (upper 100th–66th, middle 66th–33rd, lower 33rd–1st). For every k-mer associated with the primary phenotype ($n = 310{,}850$), a $2 \times 2$ contingency table summarised k-mer presence/absence in isolates within the upper and lower percentile for the secondary phenotype (Fig. 1). The genome position of k-mers significantly associated with the secondary phenotype (Fisher's exact test, P-value < 0.005) were visualised using Circos[75].

**Horizontal gene transfer among infection-associated genes**. Population genetic analyses were undertaken to compare molecular variation among 61 genes that contained infection-associated elements, correlated with a secondary infection phenotype and those that did not ($n = 1946$ genes), in asymptomatic carriage and infection isolates. For both groups, the number of alleles at each locus (determined using a whole-genome MLST approach[61] and consistency index (CI)) were calculated. The consistency of a phylogenetic tree to patterns of variation in sequence alignments was determined for each gene of interest, and constituted an inference of the minimum amount of homoplasy in these genes, as implied by the tree[76]. The CI function from the R Phangorn package[77] was used to calculate consistency indices for every single-gene alignment of the 61 genes of interest to a phylogeny constructed from a concatenated gene-by-gene alignment of 1946 genes shared by all 152 isolates used in the GWAS. The average CI of these shared genes was compared to that of the 61 genes containing pathogenicity-associated elements and correlated with secondary in vitro phenotypes.

**Risk calculation**. Pathogenicity is a complex multifactorial property. By training a classifier using the output of the GWAS analysis, we were able to go from observations of sequence variation among infection and carriage isolates to predicting phenotype and allowing risk calculation for different genotypes. To capture the non-linear and potentially complex association between sequence variation and phenotype, a Random Forest (RF) classifier was used[78]. To limit the complexity of the model, a feature selection procedure was applied. The data contained 415 isolates (141 asymptomatic, 274 infection). The set of candidate predictors consisted of 310,850 presence/absence patterns of disease-associated k-mers identified in the primary GWAS analysis and 23,561 presence/absence patterns of disease-associated and lab phenotype-correlated k-mers (Fig. 1). After filtering out the non-unique k-mer

patterns, this corresponded to 1900 and 293 predictors, respectively. In separate RF runs, the classifiers were trained using all 1900 or 293 predictors. The importance of the predictors was estimated using the built-in criterion of the RF model. The predictors were then sorted from the most to the least important. To reduce the model complexity and thereby the risk of overfitting, we applied a two-step feature selection approach. In the first step, we made use of prior biological knowledge and focused on k-mers that were correlated with known pathogenicity-associated laboratory phenotypes. In the second step, we used a data-driven procedure to pick out a small subset of the most informative predictors discovered during the first step. To evaluate the performance of models including only a small subset of the predictors, the classification accuracy of RF models including only the $l$ highest ranked predictors ($l = 1$, …$n$) was estimated using two-fold cross-validation (100 iterations).

The accuracy of the classifier was estimated by out-of-bag prediction, which gives an unbiased estimate of the out-of-sample accuracy without requiring a separate test set. The procedure exploits the subsampling step used during training where the out-of-bag prediction of isolate A is the mean prediction averaged over all trees that did not have isolate A included in their bootstrap training sample.

**Ethics**. Volunteers who donated blood for this study gave their consent as part of a research project that has been assessed by the local Human Tissue Act committee (Wales REC 6) at the Swansea University Medical School (ref: #13/WA/0190).

## Data availability

All scripts and example input and output files are available on: https://github.com/shepardlab/pGWAS and Figshare. Short-read sequence data for all 241 isolates sequenced in this study are deposited in the SRA and can be found associated with BioProject PRNJA433155. Assembled genomes are also available on figshare. NCBI genome accession numbers for isolates in the validation dataset are included in Supplementary Data 7.

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

## Acknowledgements

This work was supported by Medical Research Council (MRC) grants MR/L015080/1, MR/M501608/1 and G0081929, Biotechnology and Biological Sciences Research Council (BBSRC) grant BB/I02464X/1 and the Wellcome Trust. G.M. was supported by a Health Research Fellowship (HF-14-13) awarded by the National Institute for Social Care and Health Research (NISCHR). K.Y. was supported by a JSPS Research Fellowship for Young Scientists. J.C. was supported by the ERC grant no. 742158. H.R. was supported by funding from Damp-Stiftung. We are very grateful to Thomas Wilkinson (Swansea University) for help and advice in the IL-8 experiments, and to Jane Mikhail and Llinos Harris (Swansea University) for valuable technical input. Computational calculations were performed with HPC Wales (UK) and MRC CLIMB.

## Author contributions

S.S., G.M. and L.M. conceived the study and designed the experiments. L.M., G.M., B.P., V.P., M.Mo., H.L., M.Mi., T.M., H.R. and D.M. sampled isolates. L.M., G.M., B.P., S.S, N.K., P.T., S.L., M.L., R.B. and R.M. carried out Laboratory work. K.J., J.B., M.C.J.M. and S.S. supported data archiving. G.M., L.M., K.Y., X.D., J.P. and J.C. analysed the data. E.F., D.M., R.M., G.M. and J.C. contributed to data interpretation. S.S. L.M. and G.M. wrote the paper.

## Additional information

**Competing interests:** The authors declare no competing interests.

