## [Peer Review File · Nature Communications]

Reviewers' comments:

Reviewer #1 (Remarks to the Author):

Meric, Mageiros, et al. is a strong manuscript applying novel methods to an interesting question regarding the use of genomics to determine pathogenicity in bacteria – the results of which are also very successful in predicting the phenotype.

Overall, I think the manuscript (especially the analyses and results) is very strong. My main request is if the authors could try to maybe help clarify the analysis strategy a bit more for the reader. Box 1 is helpful, however I think a more clear flow chart of each step of the analysis, and the number of k-mers/genes identified at each stage, would improve the clarity. This is also perhaps true for the results description as well – for instance the 293 number of k-mer patterns requires jumping back to methods rather than being clear from results alone. I really like methods used here (e.g.g the use of correlated phenotypes), so think just allowing the reader to appreciate these more this would help.

Other minor points for consideration (and not required to be answered directly) are:

-Would it be possible to do an analysis of the additive effects of the k-variants for comparison with the Random Forest methods. I think this would be useful to give the result context. Also having an idea of how much the secondary phenotypes themselves can be used to predict outcome would be useful for comparison.

-Just a thought on the number of unique alleles seen in the infection vs. non infection genes- could one test to see if these unique alleles are disproportionately, say, predicted to be deleterious? This might give some additional support that the infectious samples are not just in some way unfit or carrying more rare mutations for some other reason.

Reviewer #2 (Remarks to the Author):

The authors tackle the question whether all *S. epidermidis* strains (*S. epi.*) “are created equal” in terms of their pathogenic potential. If this was the case, the species would be a true opportunistic pathogen - case B in the authors' systematics as depicted in fig. 4 of this manuscript - and the host factors would be the sole decisive determinants of pathogenicity. If not, variability of the microbial species has to be considered, possibly in two different ways: Case A: There are clones of different virulence; case C: the multifactorial pathogenicity potential is mainly driven by virulence genes, resp. their allelic variants, which are reassorted in clones with different genetic background due to horizontal gene transfer (HGT). Since *S. epidermidis* is a ubiquitous commensal, in all three scenarios, the pathogenic clones would be a subset of the colonizing clones.

To address this, the authors combine genome-wide association studies in a collection of *S. epi.* clones, which showed commensal or pathogenic behavior in their respective hosts, with laboratory determinations of traits known – or assumed – to confer increased virulence. They show that a genetic risk score comprising merely 4 short gene sequences predicts the *in vivo* behavior of the clones with 80% accuracy. This clearly refutes hypothesis B (true opportunism) and makes a case for scenario C (divided genomes).

This intriguing manuscript is of high clinical relevance. *S. epi.* is frequently found in the context of bacterial infection, however, its causative role remains undetermined in all too many cases, because it is difficult to distinguish infection from contamination by *S. epi.*, which is ubiquitous on the body surfaces. To be able to identify *S. epi.* strains of high pathogenicity would, hence, greatly benefit diagnostics and alert clinicians to the threat imposed by *S. epi.*, which is probably currently underestimated. Moreover, the insight provided into the evolutionary and molecular mechanisms

underlying pathogenicity of *S. epi.* represents a significant advance to the field of knowledge, which is of interest to a wide readership. Moreover, it is appreciated that the genome data have been made publicly available for further analysis (assuming that metadata of the clinical phenotype are contained in the public database). The paper is clearly written, and the well-designed graphics render the abstract concepts accessible.

However, in my opinion, the paper could be much stronger, if the authors widened their perspective and made full use of their unbiased approach rather than restricting the analysis to pre-assumed *in vitro* correlates of *S. epi.* pathogenicity.

Major comments

1. A major criticism concerns the selection of *S. epi.* strains for analysis. While the infectious strains are of diverse origin in terms of geographical region and time of sampling, all commensal isolates have been obtained in 2012 in the Swansea area. This deprives the authors of the possibility to meaningfully test hypothesis A, namely, the existence of clonal lineages of increased pathogenic potential. In the present strain collection, differences in clonal composition between commensal and pathogenic isolates may be due to differences in geographical distribution as well as to differences in pathogenicity. The authors have to clearly explain to the readers that their study and its results do not address scenario A and by no means exclude it.

2. In order to discriminate between the cases B and C, the authors paired *S. epi.* clones of similar genetic background but differing in their interaction with the host (commensal vs infectious). They find numerous gene sequence stretches (k-mers) differing significantly between the two phenotypic groups (Box 1; I take it that the GWAS association scores depict the real data). However, in their analysis the authors seem to ignore the majority of the most promising discriminating k-mers, e.g., all black dots above the threshold of significance in box 1, to focus on only those k-mers associated with laboratory traits of virulence (colored dots), which in comparison are lacking in discriminatory power. Hence, the authors do not make optimal use of the unbiased approach, namely to discover the most powerful biomarkers as well as novel virulence factors. Moreover, many of the ignored genetic determinants are located on plasmids which could significantly strengthen the case the authors intend to make (scenario C).

3. Rather, the authors concentrate on the co-variance of GWAS hits with established *in vitro* phenotypes that are generally assumed to indicate pathogenicity. By selecting only those k-mers correlated with these *in vitro* phenotypes, they cement the notion that these phenotypes are important in the sense of a self-fulfilling prophecy, rather than using their unbiased study's potential to put these dogmata to the test. In fact, with the exception of methicillin resistance, the correlation of the *in vitro* traits with commensal vs infectious behavior is surprisingly poor, and in the case of biofilm formation and cell toxicity even counterintuitive, as shown in Fig. 2A. The paper would be much stronger, if the authors made use of their data's full potential.

4. The authors focus on k-mers significantly co-associated with both *in vivo* and *in vitro* traits of the *S. epi.* isolates. Conversely, how many k-mers are associated with the *in vitro* traits but not with either commensal or pathogenic behavior?

5. I am quite prepared to accept that the selected *in vitro* traits (except for IL-8 induction in HaCaT) may be correlates of *S. epi.* pathogenicity in certain contexts. However, the extreme variance of these features in both asymptomatic carriage and infection questions generalizability. In fact, opposite behavior of *S. epi.* may promote infectivity and acute infection on the one hand, and adaptability and chronic infection on the other. It is not clear to me, whether the data sets used for establishing and testing the risk score are independent. Tests in independent strain collections are mandatory to establish the general validity of the score or (especially when considering the limitations outlined in comment 1) or, alternatively, refine it for specific clinical scenarios.

6. Information about the clinical characteristics of the studied strains has to be included in table S1 (e.g., asymptomatic carriage, type of infection).

Minor comments

7. Line 333 – what are the 293 k-mer patterns based on? Are these patterns of co-variance with the in vitro traits?

8. Assuming that HGT is important, it would be of interest when this occurs. Is it part of pre-adaptation causing variance within the colonizing *S. epi.* clade, or will it rather happen during chronic infection reflecting bacterial adaptation to the secondary niche?

9. The secondary niche is discussed as a dead end for infectious agents. While this is plausible in acute infection resolved by sterilizing cure or death of the host, the secondary niche could represent a reservoir for re-colonization of the primary niche in chronic infection.

Reviewer #3 (Remarks to the Author):

The manuscript “A pathogen in plain sight: disease-causing genotypes of the commensal skin bacterium *Staphylococcus epidermidis*” combines a genome-wide association approach for disease status with laboratory characteristics indicating potential pathogenicity to build a model to identify risk phenotypes.

I enjoyed reading this manuscript, as it is well-structured and has clear potential to be useful in clinical practice. I have only minor comments to make.

- Lines 262-270: You might want to check the referencing to Figure 1A and B, I think you might have switched this round, as 1A does not show the phylogenetic tree as stated in lines 264f.
- Lines 272-283: Have you tried running the two datasets together to increase statistical power? Have you considered to control for population structure in a different way, e.g. using a linear mixed model, rather than selecting pairs of isolates?
- Lines 344-357: It would help if you stated which parts of the SCCmec element were significantly associated, and also a reference to Table 1 is missing or would indeed be helpful. I am also wondering whether the authors can comment on the obvious similarity between *S. epidermidis* and *Staphylococcus aureus* in that methicillin resistance is a major contributor in the latter to its epidemic potential. Also in *S. aureus*, methicillin resistance goes hand in hand with fluoroquinolone/ciprofloxacin resistance. I noticed in Table 1 that *gyrB* comes up as a significant hit – Is this potentially related? Are the SNPs in that gene in any way similar to those that contribute to ciprofloxacin resistance in *S. aureus*? If that is the case it may be worth checking for mutations in *gyrA* as well as *griAB*, even if they didn't come up as significant. Is anything known about the ciprofloxacin susceptibility of these isolates? I think these points would be worth pointing out and discussing in either the results or the discussion section as well, as staphylococcal researchers will surely wonder about it.
- Also Lines 344-357: Out of curiosity, how much can you predict with the MEC predictor alone? Might be an interesting point to add, since it has such a large weight. I like the idea of having some gene to test for in the clinic to establish an individual risk score, and appreciate that more than one factor is contributing for sure, but if methicillin resistance was such a big contributor it would simplify things.
- Lines 423-434, especially 430: This needs a reference to Figure 1C as well. It might be worth putting down numbers for how many kmer associations were in the core and how many were in the accessory genome. Might also need a reference to Table 1 (see point below).
- Lines 460-468: You might want to add the point of MEC being the best or strongest predictor here as well.
- Figure 1: A/B part switch. C – I feel like a scale bar for strength of the hits is missing, or an indication that each ring corresponds to something.
- Figure 2: I would suggest ordering part F to match the order of A-E, to make it easier for the reader. I am not sure whether there was a reason for ordering it either way, but I would make it consistent.

- Figure 3: Has A and B parts, but not mentioned in the legend.
- Figure S4: It might be useful to put the 0Mb at 12 o'clock similar to Figure 1C. Reference to Figure 1D which does not exist? "Blips" on pangenome annotation – I am assuming these refer to GWAS spikes/hits, but this is unexplained in the legend.
- Table 1: I feel like you could make more of this, I don't think it is even mentioned in the text very much. I think it would help to state in the text how many of the main associations were in the core genome as opposed to the accessory genome, this might help solidify your claim that HGT is an important contributor. I don't think an actual number is mentioned in the manuscript so far.

RESPONSE TO REVIEWERS:

Reviewer #1 (Remarks to the Author):

1. Meric, Mageiros, et al. is a strong manuscript applying novel methods to an interesting question regarding the use of genomics to determine pathogenicity in bacteria – the results of which are also very successful in predicting the phenotype. Overall, I think the manuscript (especially the analyses and results) is very strong.

- *We are grateful for the positive comments on our work.*

2.....My main request is if the authors could try to maybe help clarify the analysis strategy a bit more for the reader. Box 1 is helpful, however a think a more clear flow chart of each step of the analysis, and the number of k-mers/genes identified at each stage, would improve the clarity. This is also perhaps true for the results description as well – for instance the 293 number of k-mer patterns requires jumping back to methods rather than being clear from results alone. I really like methods used here (e.g.g the use of correlated phenotypes), so think just allowing the reader to appreciate these more this would help.

- *Consistent with the reviewers comments, Box 1 has been revised and clarified to show the number of k-mers/genes identified at each analysis stage.*

3. Other minor points for consideration (and not required to be answered directly) are:

Would it be possible to do an analysis of the additive effects of the k-variants for comparison with the Random Forest methods. I think this would be useful to give the result context. Also having an idea of how much the secondary phenotypes themselves can be used to predict outcome would be useful for comparison.

- *Consistent with this, Figure S5 shows the segregation of infection associated k-mers in carriage and infection isolates that were not included in the GWAS. This implies additive effects in the RF approach are necessary for quantitative analysis. Consistent with the power of the MEC predictor, infection isolates show increased resistance in the phenotype assays. However, the incomplete understanding of pathogenicity means that it is possible that a given phenotype may promote opposite outcomes, for example infectivity (and acute infection) on one hand and adaptability (chronic infection) on the other. This has been discussed in the revised text (Lines 178-184, Refs 107-108).*

4. Just a thought on the number of unique alleles seen in the infection vs. non infection genes- could one test to see if these unique alleles are disproportionately, say, predicted to be deleterious? This might give some additional support that the infectious samples are not just in some way unfit or carrying more rare mutations for some other reason.

- *We welcome this comment from the reviewer, as this is something we have considered. There was a subtle increase in the average allelic variation among genes associated with infection (Figure 3A). This might be expected because of the accumulation of deleterious mutations associated with bacterial range expansion (Bosshard L, et al. 2017, Genetics 207: 669–684). However, the difference was not statistically significant and there was no observable reduction in growth rate of infection strains in the laboratory. Rather than making predictions of deleterious mutations based upon the genomes, ongoing work in the laboratory is investigating the functional consequences of infection associated variations and their impact upon fitness. The text has been revised to place our findings in the context of the reviewer's suggestion (Lines 348-351) although no additional findings have been inferred at this time.*

Reviewer #2 (Remarks to the Author):

This intriguing manuscript is of high clinical relevance. *S. epi.* is frequently found in the context of bacterial infection, however, its causative role remains undetermined in all too many cases, because it is difficult to distinguish infection from contamination by *S. epi.*, which is ubiquitous on the body surfaces. To be able to identify *S. epi.* strains of high pathogenicity would, hence, greatly benefit diagnostics and alert clinicians to the threat imposed by *S. epi.*, which is probably currently underestimated. Moreover, the insight provided into the evolutionary and molecular mechanisms underlying pathogenicity of *S. epi.* represents a significant advance to the field of knowledge, which is of interest to a wide readership. Moreover, it is appreciated that the genome data have been made publicly available for further analysis (assuming that metadata of the clinical phenotype are contained in the public database). The paper is clearly written, and the well-designed graphics render the abstract concepts accessible.

- We thank the reviewer for their positive comments. We are particularly gratified that they highlight the importance of S. epidermidis in bacterial infection, the clinical relevance, and consider our evolutionary approach to be a significant advance to the field that will be of broad interest.

However, in my opinion, the paper could be much stronger, if the authors widened their perspective and made full use of their unbiased approach rather than restricting the analysis to pre-assumed in vitro correlates of *S. epi.* pathogenicity.

- We respond to this with the related comment 2 below.

Major comments

1. A major criticism concerns the selection of *S. epi.* strains for analysis. While the infectious strains are of diverse origin in terms of geographical region and time of sampling, all commensal isolates have been obtained in 2012 in the Swansea area. This deprives the authors of the possibility to meaningfully test hypothesis A, namely, the existence of clonal lineages of increased pathogenic potential. In the present strain collection, differences in clonal composition between commensal and pathogenic isolates may be due to differences in geographical distribution as well as to differences in pathogenicity. The authors have to clearly explain to the readers that their study and its results do not address scenario A and by no means exclude it.

- It appears that we were not clear enough in the text about the origin of the isolates and have revised it accordingly (Lines 115-116). In fact the asymptomatic carriage isolate collection came from carriers in hospitals and the community from 11 countries in 3 continents (with 57/141, or 40% of all asymptomatic isolates from the UK) (Table S1).

- It is not our intention to exclude the possibility that clonal lineages with enhanced pathogenic potential may exist somewhere, or emerge in the future. The text has been revised as suggested to clarify this (Lines 285-293). However, model of disease causing clones is not consistent with the available genomic data for this species. As shown in Figure 1B, known genetic diversity in this species is captured by our sampling consistent with previous studies, with isolates from all major phylogenetic groups being represented in both the asymptomatic and the infection isolate collections.

2. In order to discriminate between the cases B and C, the authors paired *S. epi.* clones of similar genetic background but differing in their interaction with the host (commensal vs infectious). They find numerous gene sequence stretches (k-mers) differing significantly between the two phenotypic groups (Box 1; I take it that the GWAS association scores depict the real data). However, in their analysis the authors seem to ignore the majority of the most promising discriminating k-mers, e.g., all black dots above the threshold of significance in

box 1, to focus on only those k-mers associated with laboratory traits of virulence (colored dots), which in comparison are lacking in discriminatory power. Hence, the authors do not make optimal use of the unbiased approach, namely to discover the most powerful biomarkers as well as novel virulence factors. Moreover, many of the ignored genetic determinants are located on plasmids which could significantly strengthen the case the authors intend to make (scenario C).

*- We appreciate and agree that, from a clinical perspective, microbial GWAS offers great potential for significantly improving understanding of infection. As suggested we have added RF analysis results for all k-mers from the primary GWAS (Lines 246-257, 363-372, Revised Box 1, figure 2G) and emphasized that all GWAS hits are potential discriminators (314-318). However, the GWAS field is dominated by contributions from statistical geneticists, with ever increasing isolate genome collections and analysis tweaks that seldom investigate the effect of genomic associations beyond inference of putative gene function from gene ontologies without recourse to microbiology. Such studies are, in some ways, divorced from the decades of rigorous lab work by microbiologists across the world. A central tenet in this study is that by bringing microbiology, genomics and bioinformatics together we come closer to a genuine understanding of the evolutionary processes underlying infection by *S. epidermidis* and provide a road map for detailed characterization of how genotype variation influences clinically relevant phenotypes in pathogens where current knowledge is incomplete. Notwithstanding this, we appreciate that the interpretation of the significance of laboratory phenotypes builds on 'pre-assumed' literature on clinically relevant in vitro phenotypes, and cite numerous influential studies (especially Otto, Rohde). However, rather than completely change the nature of the paper we have preferred to add additional RF results (as suggested), emphasize the utility of the primary GWAS as markers of infection and explain the importance of phenotype correlation in the revised submission, consistent with the reviewers comments (Lines 308-309, 384-391). See also the response to comment 5b (iv) below.*

3. Rather, the authors concentrate on the co-variance of GWAS hits with established in vitro phenotypes that are generally assumed to indicate pathogenicity. By selecting only those k-mers correlated with these in vitro phenotypes, they cement the notion that these phenotypes are important in the sense of a self-fulfilling prophecy, rather than using their unbiased study's potential to put these dogmata to the test. In fact, with the exception of methicillin resistance, the correlation of the in vitro traits with commensal vs infectious behavior is surprisingly poor, and in the case of biofilm formation and cell toxicity even counterintuitive, as shown in Fig. 2A. The paper would be much stronger, if the authors made use of their data's full potential.

- As described above, RF of the primary GWAS is now included (Lines 247-257, 364-374, Revised Box 1, figure 2G). The power of Random Forest approaches is diminished if there are too many highly correlated predictors in the procedure. Having said this, we have carried out an RF on the primary GWAS output as suggested. This highlights the importance of SCCmec elements as predictors (Figure 2G). To reduce the model complexity and thereby the risk of overfitting, we applied a two-step feature selection approach. In the first step, we made use of prior biological knowledge and focused on k-mers that were correlated with known pathogenicity associated laboratory phenotypes. In the second step, we used a data-driven procedure to pick out a small subset of the most informative predictors discovered during the first step. The text has been amended to describe this additional analysis and its limitations (Lines 247-257, 364-374, Revised Box 1, figure 2G)

4. The authors focus on k-mers significantly co-associated with both in vivo and in vitro traits of the *S. epi.* isolates. Conversely, how many k-mers are associated with the in vitro traits but not with either commensal or pathogenic behavior?

- *Only k-mers that were associated in the primary GWAS (in vivo) experiment were correlated with in vitro phenotypes. Box 1 has been revised to make this more clear. The number of k-mers from the primary GWAS that did not correlate with the laboratory phenotypes is mentioned in the revised text (Line 335).*

5(a) I am quite prepared to accept that the selected in vitro traits (except for IL-8 induction in HaCaT) may be correlates of *S. epi.* pathogenicity in certain contexts. However, the extreme variance of these features in both asymptomatic carriage and infection questions generalizability. In fact, opposite behavior of *S. epi.* may promote infectivity and acute infection on the one hand, and adaptability and chronic infection on the other.

- *We are grateful for this input and consistent with the reviewer's suggestion we have amended the text and added additional references addressing these points. We retain the data from the phenotype experiments as a central aim is to enhance understanding of the evolution of pathogenicity with the phenotype linked GWAS approach. However, the revised manuscript describes the potential of unlinked GWAS hits as potential biomarkers (independent of phenotype)(Lines 308-309, 391-396) and emphasizes that because of the principal adaptation to the commensal niche and the absence clear discriminatory phenotypes (see also response to 5b below) it is possible that a given phenotype may promote opposite outcomes, for example infectivity (and acute infection) on one hand and adaptability (chronic infection) on the other. The text has been amended to reflect this (Lines 178-184).*

5(b) It is not clear to me, whether the data sets used for establishing and testing the risk score are independent. Tests in independent strain collections are mandatory to establish the general validity of the score or (especially when considering the limitations outlined in comment 1) or, alternatively, refine it for specific clinical scenarios.

- *Consistent with the reviewer's comments the manuscript has been revised as follows:*

(i) *We have made it clear that the isolate collection represents known diversity within *S. epidermidis* (Lines 99, 179-181, 284-288, 290-293). This is relevant to point (ii) below.*

(ii) *We have revised the text to describe how evaluation of the risk score is done by cross-validation, which is the standard technique for measuring the predictive power of a classifier and how well the classifier will generalize to an independent dataset. Random forests have a built-in procedure for this, known as out-of-bag estimation, which removes the need for traditional cross-validation where the data is explicitly split into a training and test set. The procedure exploits the subsampling step used during training. Basically, the out-of-bag prediction of isolate A is the mean prediction averaged over all trees that did not have isolate A included in their bootstrap training sample. Consequently, random forests don't require a separate test set to give an unbiased estimate of the test set error. This has been explained, with relevant referencing (Ref 109), in the revised manuscript (Lines 262-266).*

(iii) *The greatest challenge for risk prediction based upon infection associated k-mers is that samples from asymptomatic carriage may include strains that have the potential to cause infection later, after our samples were taken. This depends on the opportunity to infect, specifically the healthcare related procedures a person will be subjected to. Thus, while it is relatively straight-forward to obtain isolates from confirmed infection, it is nearly impossible to get a representative sample of carriage strains that does not contain isolates with the potential to cause disease. A practical solution is to increase the number of samples and thus the statistical power to detect associations. We aimed to provide the proof of concept and*

describe a way forward for clinical assessment as every larger databases are compiled. The text has been amended to clarify this (Lines 405-411, 545-547).

(iv) In light of the above, it may seem surprising quite how powerful a single predictor can be and we fully appreciate how the potential to use infection biomarkers (Reviewer 2, comment 2), all be it just one component of this study, will be of interest to public health microbiologists. As suggested by the reviewer, we have carried out a validation of the best RF predictor k-mer (in MecA) on another dataset available on NCBI. The dataset (new Table S6) is small because few confirmed asymptomatic carriage isolates have been sequenced outside of this study. The prediction from the validation was 67% consistent with the 75% accuracy on the large primary dataset. The manuscript has amended to include this (Lines 411-417).

6. Information about the clinical characteristics of the studied strains has to be included in table S1 (e.g., asymptomatic carriage, type of infection).

- All the available isolate metadata is included in table S1.

Minor comments

7. Line 333 – what are the 293 k-mer patterns based on? Are these patterns of co-variance with the in vitro traits?

- These are presence absence patterns for the k-mers. This has been clarified in the revised text (Lines 369-370)

8. Assuming that HGT is important, it would be of interest when this occurs. Is it part of pre-adaptation causing variance within the colonizing *S. epi.* clade, or will it rather happen during chronic infection reflecting bacterial adaptation to the secondary niche?

*- It is difficult to pinpoint the exact location where HGT occurred as we are essentially recording realized recombination, ie recombination that was beneficial (or at least not detrimental) and has stuck in the population. From a theoretical evolutionary perspective, HGT benefits by allowing the spread of adaptive genes in heterogeneous environments (commensal and invasive niches). In a genetically diverse community of commensal *S. epidermidis* HGT would promote: (i) the emergence of lineages (at the boundary between niches) that could colonize the new niche effectively and (ii) ongoing adaptation as positively selected genes that confer an advantage in the invasive niche, sweep through the invasive population. This has been described in the revised text (Lines 499-512).*

9. The secondary niche is discussed as a dead end for infectious agents. While this is plausible in acute infection resolved by sterilizing cure or death of the host, the secondary niche could represent a reservoir for re-colonization of the primary niche in chronic infection.

- The text has been amended consistent with the reviewers comment (lines 476-477)

Reviewer #3 (Remarks to the Author):

1. The manuscript “A pathogen in plain sight: disease-causing genotypes of the commensal skin bacterium *Staphylococcus epidermidis*” combines a genome-wide association approach for disease status with laboratory characteristics indicating potential pathogenicity to build a model to identify risk phenotypes. I enjoyed reading this manuscript, as it is well-structured and has clear potential to be useful in clinical practice. I have only minor comments to make.

- We are pleased that the reviewer considers our work potentially useful in clinical practice as taking GWAS from in silico studies to real microbiology is a central aim for us.

2. Lines 262-270: You might want to check the referencing to Figure 1A and B, I think you might have switched this round, as 1A does not show the phylogenetic tree as stated in lines 264f.

- *The figure referencing typos have been corrected*

3. Lines 272-283: Have you tried running the two datasets together to increase statistical power? Have you considered to control for population structure in a different way, e.g. using a linear mixed model, rather than selecting pairs of isolates?

- *GWAS approaches that use a linear fixed effects or a mixed model have been developed for bacteria by Jukka Corander and colleagues (Lees et al, 2016 Nat Commun 7: 12797; Lees et al, 2018 Bioinformatics). This is a highly scalable approach has considerable benefit for analysing 1000s of genomes from highly diverse datasets. However, the k-mer approach employed in this study is also widely used (Sheppard SK et al 2013, PNAS 110, 11923-11927; Pascoe B et al 2015, Environ Microbiol 17, 4779-4789; Yahara K et al 2017, Environ Microbiol 19, 361-380) and has practical advantages. Specifically, the approach in this study determines the significance of associations by comparison to expectation based on the clonal frame of the population (the tree). As closely related (genetically) isolates were found to have divergent phenotypes (infection vs carriage), the most targeted and rigorous method to identify associated elements was to run the test using the tree. By highlighting the significance of associations identified in this way, found in multiple pairs, we greatly reduced the possibility of interference from SNPs that were inherited by chance, as it is highly unlikely that these would be inherited across the tree only in the infection strains (and not their clonally matched counterparts).*

3. Lines 344-357: It would help if you stated which parts of the SCCmec element were significantly associated, and also a reference to Table 1 is missing or would indeed be helpful.

- *As suggested, we now describe which of the phenotype correlated GWAS hits are associated with SCCmec and reference Table 1 (lines 391-396).*

4. I am also wondering whether the authors can comment on the obvious similarity between *S. epidermidis* and *Staphylococcus aureus* in that methicillin resistance is a major contributor in the latter to its epidemic potential. Also in *S. aureus*, methicillin resistance goes hand in hand with fluoroquinolone/ciprofloxacin resistance. I noticed in Table 1 that *gyrB* comes up as a significant hit – Is this potentially related? Are the SNPs in that gene in any way similar to those that contribute to ciprofloxacin resistance in *S. aureus*? If that is the case it may be worth checking for mutations in *gyrA* as well as *grlAB*, even if they didn't come up as significant. Is anything known about the ciprofloxacin susceptibility of these isolates? I think these points would be worth pointing out and discussing in either the results or the discussion section as well, as staphylococcal researchers will surely wonder about it.

- *As suggested, we now discuss similarities in the role of methicillin resistance *S. epidermidis* and *S. aureus* epidemic potential, describe the correlation with fluoroquinolone resistance (highlighting *gyrB* in Table 1) and contrast the epidemic spread of a discrete number of MRSA clones with the emergence of multiple disease causing *S. epidermidis* clones distributed across the phylogeny (lines 532-544, refs 102-105). We welcome the suggestion to look in more detail at the SNPs in *gyrA/B* and *grlA/B*, about which little is known on the population scale in *S. epidermidis* (including the isolates in this study). There are >23K k-mers, in 61 genes, that are associated with infection in the phenotype correlated dataset, and these elements clearly contain important information about the genomic changes associated*

with invasive strains and future work, informed by this study, will aim to further characterize the functional consequences, and covariance, of disease associated SNPs.

5. Also Lines 344-357: Out of curiosity, how much can you predict with the MEC predictor alone? Might be an interesting point to add, since it has such a large weight. I like the idea of having some gene to test for in the clinic to establish an individual risk score, and appreciate that more than one factor is contributing for sure, but if methicillin resistance was such a big contributor it would simplify things.

- The text now reads ‘...the highest ranked MEC associated predictor reached a classification accuracy around 75% on its own, potentially offering a very simple target for clinical investigation of S. epidermidis risk’ (lines 379-382).

6. Lines 423-434, especially 430: This needs a reference to Figure 1C as well. It might be worth putting down numbers for how many kmer associations were in the core and how many were in the accessory genome. Might also need a reference to Table 1 (see point below).

- Consistent with the reviewers comments the figures have been referenced. Also, the number of GWAS k-mer associations in 250 core and 386 accessory genes are given in Table S2.

7. Lines 460-468: You might want to add the point of MEC being the best or strongest predictor here as well.

- As suggested, this is now made clear in the text (lines 379-382, 532-535).

8. Figure 1: A/B part switch. C – I feel like a scale bar for strength of the hits is missing, or an indication that each ring corresponds to something.

- We have corrected the text so that Figure 1A and B are referred to correctly. The legend for Figure 1C has been amended to improve clarity and describe the significance of the concentric rings.

9. Figure 2: I would suggest ordering part F to match the order of A-E, to make it easier for the reader. I am not sure whether there was a reason for ordering it either way, but I would make it consistent.

- The order of the plots in panel F has been changed to match the order of A-E on the revised Figure 2.

10. Figure 3: Has A and B parts, but not mentioned in the legend.

- Panels A and B are described in the revised legend.

11. Figure S4: It might be useful to put the 0Mb at 12 o’clock similar to Figure 1C. Reference to Figure 1D which does not exist? “Blips” on pangenome annotation – I am assuming these refer to GWAS spikes/hits, but this is unexplained in the legend.

- The orientation of the pan genome representation has been changed as suggested, with 0Mb at 12 o’clock. Also, the typo referring to Figure 1D in the legend of Figure S4 has been corrected and now reads 1C.

12. Table 1: I feel like you could make more of this, I don’t think it is even mentioned in the text very much. I think it would help to state in the text how many of the main associations were in the core genome as opposed to the accessory genome, this might help solidify your claim that HGT is an important contributor. I don’t think an actual number is mentioned in the manuscript so far.

- As suggested, and consistent comment 6 of Reviewer 3, we now make more of the results in Table 1. Specifically: the number of core and accessory genes is now described in the text for all GWAS hits (lines 301-303); the genes associated with SCCmec are discussed (lines 391-396); Table 1 is mentioned 5 times in the revised manuscript.

REVIEWERS' COMMENTS:

Reviewer #1 (Remarks to the Author):

[No further comments for author.]

Reviewer #2 (Remarks to the Author):

**Dear authors,
thank you for the clarification!
Best regards
Barbara Bröker**

Reviewer #3 (Remarks to the Author):

The revised version of the manuscript adequately addressed all my comments, and I have no further recommendations to improve the paper.